# High-Performance Project Teams: Analysis from the Stoic Approach

Nelson Moreno-Monsalve [1], Marcela Delgado-Ortiz [1], Alfredo Sanabria-Ospino [2], Tamara Tatiana Pardo Ezcurra [3], Yoni Wildor Nicolás Rojas [4] and William Fajardo-Moreno [5,*]

[1] Faculty of Engineering, Universidad Ean, Bogotá 110221, Colombia; smdelga-do@universidadean.edu.co (M.D.-O.)

[2] Faculty of International Business, Universidad de Santo Tomas, Bucaramanga 680006, Colombia; alfredoenrique.sanabria@ustabuca.edu.co

[3] Department of Research, Universidad Privada Peruano Alemana, Lima 15064, Peru; tama-ra.pando@upal.edu.co

[4] Department of Research, ELP Escuela Superior La Pontificia, Ayacucho 06002, Peru; yoninico-las@elp.edu.pe

[5] Department of Research and Transfer, Universidad Ean, Bogotá 110221, Colombia

[*] Correspondence: wfajard74913@universidadean.edu.co; Tel.: +57-132-1240-2492

**Abstract:** Nowadays, any consideration regarding project management implies an all-encompassing approach constituted by three perspectives: organizational, human, and engineering. In this sense, it is necessary for the project manager to reflect on various issues, such as ethics, the understanding of the environment, leadership, teamwork, achieving results, change in management, the adoption of new technologies, and knowledge management, among many others, all this with the intention of ensuring success through value creation, consistent and aligned with the expectations of the different stakeholders. Because of the adoption of a comprehensive vision of project management, disciplines such as philosophy have become logical within the reasoning of organizational management. Classic postulates such as those of the Stoic school have become references for modern leader-managers because of their approach associated with personal development, self-discipline, self-control, logic, balance with the environment, and the creation of social value. The research analyzed how the postulates of the Stoic philosophy can determine the high performance of project teams. To achieve this, 70 project managers from different small and medium enterprises (SMEs) located in Bogotá, Colombia, were surveyed about the seven main thoughts of Stoicism and the possible relationship with the performance of their project teams. Based on their answers, a correlational model of structural equations was applied to obtain the conclusions of this study. The results obtained in this investigation are intended to contribute significantly to the maturity and modernization of the project management discipline.

**Keywords:** project management; stoicism; project teams; high performance; modern management

## 1. Introduction

The Stoic school was founded by Zeno of Citium in 300 BC under a rational approach framed in cause–effect; nothing happens by chance but by causality, and destiny is forged through the decisions and the passion of the human being to achieve their goals [1]. Stoic thinkers divide philosophy into three parts: logic, which defines science and knowledge; physics, which defines the interpretation of the world and things; and morality, which defines the behavior of the human being [2].

Some of the most recognized thinkers for their contribution to the development of Stoicism throughout history are Epictetus, Seneca, and Marcus Aurelius [3]. In his postulates, Epictetus recognizes philosophy as the love of wisdom and happiness because of virtue and tranquility. For Seneca, life is understood as a cycle of good and bad events in which the human being must strengthen themselves in times of abundance to be prepared

in times of crisis. For Marcus Aurelius, the search for the common good through knowledge and rational thought is what gives meaning to life, and the decisions that leaders make should be based on the common good [4].

In general, stoicism focuses on the control of emotions; the human being must strengthen his positive emotions and be in constant control of negative emotions. Human beings live according to their social and rational nature, managing to improve humanity through reason [1]. A stoic can be defined as an ideal, not a real being, who is in the permanent search for progress through continuous learning, building an objective vision of themselves and the world around them, acting rationally, mastering their impulses, and focusing their efforts on what they want. They can change by understanding how their actions determine their future [3]; that is, a stoic is a person who embraces high performance.

From the point of view of high performance, we see how the human nature of projects allows us to focus the efforts of people toward the achievement of a common goal, giving priority to actions that allow the achievement of a result in the form of a product or service [5]. In this way, the performance of the work teams of the environment in which the projects are developed, as well as the ability of people to design and materialize solutions, highlights the importance of identifying the characteristics that distinguish a work team. Efficiency, which facilitates the achievement of results, ensures the quality expected by stakeholders [6]. An incompetent work team is the worst enemy of the project; therefore, their interpersonal skills, academic training, and experience become key factors for success. Characteristics such as leadership ability, understanding of the environment, self-motivation, communication, influence, decision making, political and social awareness, and talent for negotiation are high performance qualities [7].

Thus, this research aims to contribute to the problem of project success, specifically to the problem of project failure, with team performance being one of the causes identified with the greatest impact on the results. In this sense, in accordance with the modern vision of project management, the Stoic approach provides a different alternative that can be applied to strengthen the capabilities of project teams.

Next, some principles of Stoic philosophy are described, with the subsequent purpose of determining which of them lead work teams toward high performance, evidencing the transversality of knowledge and the complementarity of the different disciplines.

## 2. Theoretical Framework

### 2.1. Stoic Principles

In Stoicism, seven postulates focused on the growth of the human being have been defined: act with virtue; control emotions; perception of reality; focus on what is important and controllable; learn from mistakes; value time; and search for the common good. Each of these claims is de-tailed below.

### 2.1.1. Act with Virtue

The Stoics define virtue as the set of necessary actions that a human being must perform to achieve excellence. In this sense, four basic virtues are defined to achieve the ideal self: wisdom, temperance, justice, and courage. In this way, they focus on the human being acting as a decision maker between good and evil. Good is to act with virtue as the main source of happiness, and evil is to be guided by ephemeral pleasures in satisfaction [4]. Wisdom is achieved by strengthening the ability to respond to the different circumstances of life through self-knowledge and the knowledge of the reality that surrounds us. This is what the Stoics call practical wisdom or prudence, which matures through experience [8].

For its part, temperance can be interpreted as discipline and self-control, that is, the personal requirement that helps the human being to have willpower so as not to give in to their impulses and instincts, being moderate and not losing their head despite the challenges of life. Temperance leads to better decisions, putting aside the ego, and focusing energy on actions with a purpose [9]. Justice focuses on fairness, on being able to help others without expecting anything in return. The reward for a good deed lies in having

done it. This Stoic postulate sets aside the personal good and focuses on the common good. People are not a means; on the contrary, they are an end. Justice privileges morality, integrity, leadership, teamwork, and good personal relationships [10].

Finally, courage is the virtue that allows you to do the right thing despite the circumstances, overcoming the obstacles that come along the way without giving up or giving up on the set purpose, without being affected by feelings such as fear, anger, or sadness that cloud your thinking and prevent you from making the right decisions. Courage must be accompanied by prudence. "If you strive to acquire knowledge (wisdom) if you treat others well (justice) if you act despite fear (courage), and if you overcome obstacles and temptations (temperance), you will surely do well in life" [9], p. 47.

### 2.1.2. Control of Emotions

For Stoics, the control of emotions is essential to achieve freedom. If you do not have control over what you think or feel, the decisions you make may be wrong, and you may also be condemned to senseless slavery. Stoic freedom is defined as the ability of a human being to act guided by reason and not by emotions or desires [11]. If it is possible to control one's mental state, better results are obtained in the efforts undertaken, creating greater value for oneself and for others. The ability to strategically renounce emotions, superficial pleasures, and trivial distractions must be strengthened. In this way, the spirit is strengthened, and the body is dominated. The closer a human being is to a calm mind, the closer he is to his inner strength [12].

### 2.1.3. Perception of Reality

The interpretation of reality can lead human beings to make wrong decisions. From the Stoic approach, it is necessary to take a prudent time to think before acting. Not necessarily, the perception of reality makes it true. The human being tends to confuse their thoughts with real events or certain things. Likewise, each person can have their own abstraction of reality, which may depend on how they understand the world, their beliefs, or their degree of knowledge. For the Stoics, the human being does not react directly to an event but to the interpretation that they make of it. First impressions should always be questioned. In this way, you can have a more productive and accurate perspective of your own, obtaining more forceful results. There is no single truth, but the truth can be reached through joint perspectives [13].

### 2.1.4. The Important and Controllable

From an efficiency perspective, the Stoic philosophy encourages the focus of efforts on what is important and can be kept under control. Caring about the intention of the actions and not the result should be the primary goal. Actions depend on the human being and on his influence; results often depend on good fortune. The human being must always give their best effort, for this is what can be controlled, without worrying about the result, which will be favorable and in accordance with the actions carried out [14]. Likewise, you must clearly define what you want to achieve and draw up a plan or a route to follow. This guarantees the focus of the actions, thus improving the probability of success. Defining what is important allows for the generation of a valuable mixture with what is controllable, ensuring, to a certain extent, the quality of the results. Time should be spent thinking about things that meet both conditions: important and controllable [15].

### 2.1.5. Learning from Mistakes

For Stoic philosophy, failure is necessary to learn. Being wrong is important because it allows you to advance on the path to excellence, strengthening virtue. The important thing is to learn from mistakes and try not to make the same mistakes again. The goal of making mistakes is to learn, change the way reality is interpreted, and strengthen the spirit. A human being who makes mistakes and learns nothing from their mistakes wastes time and moves away from excellence. The person who makes mistakes and understands

that obstacles are part of the path strengthens their spirit and has greater control over their emotions. Learning to forgive oneself is also a way of learning the limits of one's will while becoming aware of mistakes and deciding not to make them again [16].

### 2.1.6. The Value of Time

Time is a finite resource that should not be wasted on unimportant or worthless things. For the Stoics, death is a reality that leads us to think about the here and now, seeking to live a fuller life. The efforts, actions, and decisions that are taken must have time as a reference. Time is valuable and should not be wasted on projects that do not create value. Lost time cannot be recovered, but time well spent generates favorable feelings and provides a glimpse of the near future. Human beings must focus their efforts and skills on making the most their time properly, always thinking about what is important on their way to virtue. You must be efficient in time management, generating significant progress on what is important and what is under control [17].

### 2.2. High Performance

The path to virtue involves caring for others. For the Stoics, the correct selection of the people who accompany us on a day-to-day basis is an important task, because they can define our way of thinking and acting in a certain way, positively or negatively influencing our results [18]. A person with physical and mental abilities cannot refuse to hold positions of leadership or government; they would be selfish and have little compassion for people affected by social problems. If you have the skills, you must put them at the service of others in pursuit of the common good—the power to help [19].

A work team can be defined as a small group of people with complementary skills who are committed to achieve a common purpose [20]. In this sense, Rincon and Zambrano [21] mention that in a work team, everyone must contribute their individual effort to achieve the expected results. Likewise, communication must flow between all members. A work team is a source of learning in which the interaction of its members allows for continuous improvement through the transfer of experience and knowledge [22]. In accordance with the above, it can be mentioned that the high performance of a work team will depend largely on the quality of its members, represented in the skills that each one can contribute to achieve the expected results, as well as appropriate environmental conditions that encourage their personal and professional development [23]. Thus, it is essential to correctly manage the diversity of personalities, cultures, and talents to ensure the creation of value through the sum of efforts.

Gil and Alcover [24] highlight in their research that the main objective of integrating a work team is to achieve the highest degree of efficiency possible, providing a timely and high-quality response to environmental challenges. MacMillan [25] presents six elements that characterize a high-performance team:

- Common purpose: A work team can achieve exceptional performance when each of its members is committed to achieve a common goal.
- Clear roles: To achieve high performance, the roles that will be supported for developing tasks must be clearly defined. This will make it possible to specify, divide, and display the work, framing it in time. Collective effort is the key to achieving synergistic results.
- Accepted leadership: High-performing teams need clear and competent leadership. Manager incompetence is a project's worst enemy.
- Effective processes: A high-performance work team must have clear action processes that allow them to achieve the proposed objective.
- Strong relationships: The diversity of skills, experiences, and knowledge of team members strengthens relationships through constant interaction.
- Excellent communication: quick and decisive decision making is a priority when competing. Response capacity is a key factor and depends on permanent communication between team members.

Other of the most common characteristics of high-performance teams are the shared vision of reality and day-to-day life, open relationships that facilitate communication and the free expression of ideas, the definition of activities considering the relevant issues for the projects, the permanent search for excellence and value creation, clearly defined working methods, consensual decision making, and the resolution of conflicts objectively and sometimes in advance [26]. The profile of the people who make up a work team is a key factor in achieving project success. Identifying key skills of the team members facilitates the selection processes and allows efforts to be focused on issues that create value for the project. In the knowledge society, work teams are characterized by having a high level of freedom in decision making, their own interpretation of reality is respected, and the creation of value and innovation are encouraged. Common purposes that deliver value to stakeholders enhance collective participation and improve the quality of results [27]. Based on this theoretical analysis, two research hypotheses were proposed:

**H1:** *Personal characteristics such as acting with virtue, controlling emotions, and an adequate perception of reality have a positive effect on the performance of project teams.*

**H2:** *Efficiency characteristics such as focusing on what is important and controllable, learning from mistakes, and valuing time have a significant positive effect on project team performance.*

In this sense, a conceptual model was structured to represent the different variables identified in Stoic philosophy and their relationship with high performance and value creation. Figure 1 presents the proposed conceptual model; it includes the six postulates of Stoic philosophy that have been taken and grouped into two perspectives: person and efficiency. Likewise, the postulate of the search for the common good is presented transversally. Together, the seven tenets have a positive impact on high performance and lead to value creation.

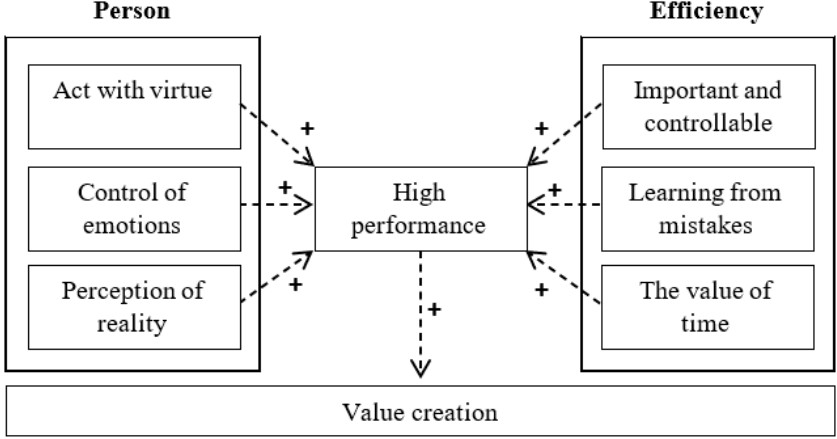

**Figure 1.** Conceptual model. Source: the authors.

## 3. Research Methodology

This research is theoretical, descriptive, and structured [28]. In the first part, an interpretive work of bibliographic sources was developed to select the variables that support the study. Based on the results obtained, 2 research hypotheses and a conceptual model that integrates them were proposed. Then, a survey was designed using the selected variables. The survey consisted of 35 questions, arranged in 6 groups, 5 for each of the 6 identified variables, and 5 more questions for the characterization of the sample. The survey was structured on a Likert scale: totally agree, agree, neither agree nor disagree, partially disagree, and totally disagree.

The validation of the survey focused on determining the validity and reliability index. The validity index was worked through Aiken's V coefficient, with the help of seven expert

researchers in human talent and high performance. The proposed questions were evaluated in three specific aspects: the clarity of writing, conceptual approach, and the relationship of the question with the variable of interest [29]. On the other hand, in the case of the reliability index, a pilot test was conducted by applying the measurement instrument to 10 project managers. With the data collected, Cronbach's alpha index was calculated, which yielded a degree of precision of the reliability instrument of 0.925.

The sample was calculated using a probabilistic method applied to a population of 79 project managers, corresponding to graduates of a master's degree in project management from Ean University, segmented from the first class of 2021. In this sense, with a confidence interval of 95% and an error rate of 4%, the sample obtained was 70 people. Table 1 shows the technical data sheet for the research.

**Table 1.** Research technical sheet.

| Characteristic | Description |
|---|---|
| Data collection period | From February to June 2021 |
| City of application | Bogotá, Colombia |
| Profile of the person interviewed | Project managers: technology, infrastructure, and services |
| Sampling method | Probability sampling |
| Confidence interval | 95% |
| Error rate | 4% |
| Sample | 70 |
| Data collection medium | Remote interview, electronic survey |

Source: the authors.

Finally, to process the data, a correlational model of structural equations was applied, which belong to the group of multivariate statistical models, and which, due to their precision, allow for the estimation of the effect and relationships between multiple variables, therefore it agrees with the purpose of this study. It was decided to apply a structural equation model for data processing because this type of statistical tools allow for greater flexibility compared to regression models, facilitating the inclusion of measurement errors in the dependent and independent variables, reducing the complexity in calculating the results.

## 4. Analysis of Results

The information collected was processed using IBM SPSS 26 software in addition to the AMOS version 24 extension. A factor analysis model was used to process the data. Table 2 shows the configuration of the variables used to obtain the correspondence factor analysis (CFA). In total, 27 observed variables belonging to seven factors were used, grouped into three latent variables.

To support the robustness and veracity of the model in the research, a confirmatory factor analysis (CFA) and a structural equation model (SEM) [30] were used. These studies provide detailed information on the estimation of the model and facilitate the analysis. between the links of the variables, allowing a holistic understanding of the theoretical framework [31]. First, the CFA was performed with the objective of validating the model and confirming the theory [32], in accordance with what is presented in Figure 2. Subsequently, to validate the research hypotheses, the SEM method is used [33]. In this way, the statistical adjustment will be calculated, as shown in Figure 3. Likewise, Table 3 shows the statistical adjustment for the CFA and SEM, where the chi-square value was 321,931 with 321,000 degrees of freedom for both measurement and structural models.



**Table 2.** Correspondence factor analysis.

| Studied Variable | Coding | Factor | Latent Variable |
|---|---|---|---|
| Virtue_P2 | V2 | | |
| Virtue_P4 | V4 | | |
| Virtue_P4 | V4_A | Act with virtue | |
| Virtue_P5 | V5 | | |
| Emotions_P2 | E2 | | |
| Emotions_P3 | E3 | | |
| Emotions_P4 | E4 | Control of emotions | Person |
| Emotions_P4 | E4_A | | |
| Emotions_P5 | E5 | | |
| Reality_P2 | R2 | | |
| Reality_P3 | R3 | Perception of reality | |
| Reality_P4 | R4 A | | |
| Imp-Cont_P2 | I2 | | |
| Imp-Cont_P4 | I4 | | |
| Imp-Cont_P4 | I4_A | Important controllable | |
| Imp-Cont_P5 | I5 | | |
| Learning_P3 | A3 | | |
| Learning_P4 | A4 | | |
| Learning_P4 | A4_A | Learning from mistakes | Efficiency |
| Learning_P5 | A5 | | |
| Time_P2 | T2 | | |
| Time_P3 | T3 | | |
| Time_P4 | T4 | Value of time | |
| Time_P4 | T4_A | | |
| Performance_P4 | D4 | | |
| Performance_P4 | D4_A | High performance | Performance |
| Performance_P5 | D5 | | |

Source: the authors, based on the data collected.

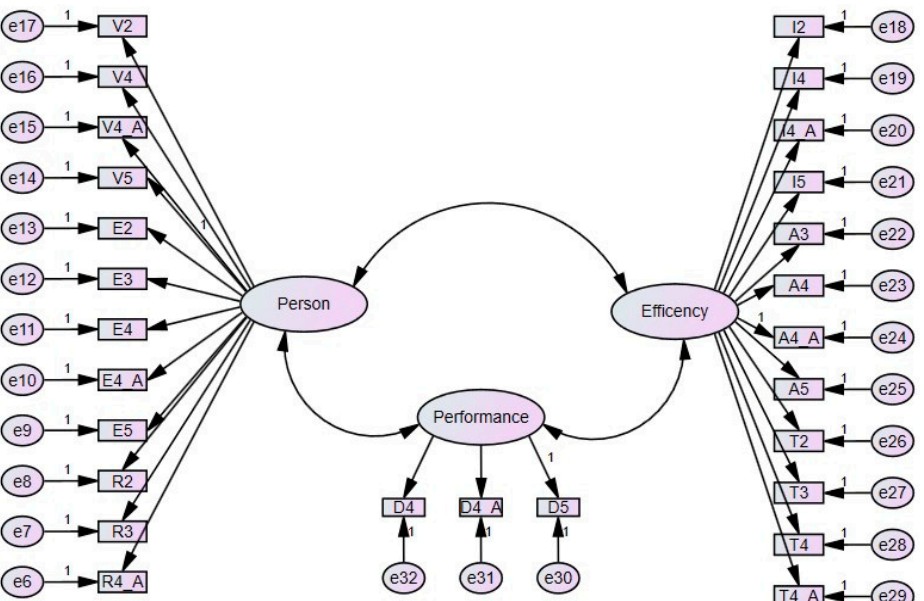

**Figure 2.** Measurement model source: the authors, based on the data collected.

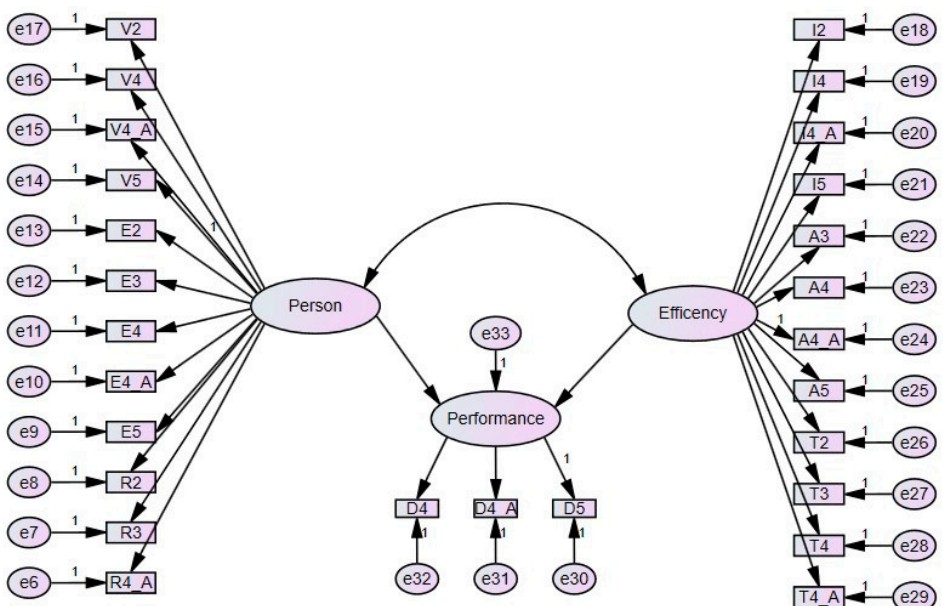

**Figure 3.** Structural model source: the authors, based on the data collected.

**Table 3.** Statistical adjustment.

| Statistical Adjustment | Interception | Values |
|---|---|---|
| χ2/df | 1–3 | 1.003 |
| CFI | >0.95 | 0.963 |
| RMSEA | <0.06 | 0.006 |
| PClose | >0.05 | 0.973 |
| TLI (Tucker–Lewis) | >0.95 | 0.959 |
| IFI (Incremental Fit Index) | >0.9 | 0.983 |
| PCFI (Comparative Fixed Parsimony Index) | >0.6 | 0.881 |

Source: the authors, based on the data collected.

As shown in Table 3, the AFE measurement model and the SEM structural model align with the information collected. The findings presented were carried out using the maximum likelihood statistical method, which allows for the testing of direct causal hypotheses [34,35]. Likewise, the structural model presented in Figure 3 evaluates the hypothetical direct causal relationship through the standardized loadings between PERSON and EFFICIENCY of hypotheses 1 and 2. The findings show that all direct relationships are significant and positive. constructs towards performance efficiency (β = 0.21; $p$ = 0.049) and person (β = 0.20; $p$ = 0.048) $p < 0.05$ two-tailed. Therefore, alternative hypotheses H1 and H2 can be accepted, concluding that personal characteristics, such as acting with virtue, controlling emotions, and an adequate perception of reality, positively impact the performance of project teams, such as focusing on what is important and controllable, learning from mistakes, and valuing time have a significant positive effect on project team performance.

## 5. Discussion

One of the first reflections of this research is how organizations are facing different environmental challenges. For example, the sustainable approaches are some of the most important for the global economy because it requires rethinking the way companies operate to achieve the balance between productivity and care of the environment [36]. In this context, approaches such as those of gymnastics companies [37] are significant, in which the effort is focused on achieving results, leaving aside the process, making organizational structures more flexible, and developing the capabilities of human talent. Likewise, a

positive trend is observed for organizational strategies to be characterized by their flexibility and ambidexterity, prioritizing the optimization of processes without neglecting the near future [38]. Therefore, the development of human talent capabilities is one of the key success factors that sustain the competitive advantage [39]. Strengthening the capabilities of work teams will significantly impact the projects that frame organizational strategy.

Second, project management requires work teams to have a mix of hard and soft skills that allow them to achieve the objectives set and create value for stakeholders [40]. Soft skills regulate personal behavior and your interaction with other people, whereas hard skills focus on the technical aspects necessary to perform the job efficiently [41]. Thus, in the field of project management, the same importance is observed between hard and soft skills. In multidisciplinary engineering teams, it is important that the members not only have technical knowledge but also the ease of interacting with other people to achieve the common objective [42]. Therefore, a positive evaluation is observed on the level of development of employees' skills and its impact on the performance of organizations [43], even more so when organizational strategies are achieved through the projects executed [44]. Under this reflection, we see how the principles of the Stoic approach, grouped into personal and efficiency as presented in this research, make special sense within the framework of the soft and hard skills that a high-performance project team should have.

Every project executed must have the objective of creating value for stakeholders. Thus, efforts must focus on satisfying the needs of the communities, creating the greatest possible value, without negatively or disproportionately affecting the environment. Therefore, project managers must achieve the expected economic development without risking the future of the next generation. Ultimately, sustainable development must be achieved that meets current needs without compromising future capabilities.

Finally, it is important to mention that the main limitation of this study corresponds to the number of individuals that make up the sample, this could statistically bias the results obtained, however, the experience of the project managers surveyed adds confidence to the results provided.

## 6. Conclusions

Within the modern project management approach, measuring success through the generation of value is disruptive and places the project management discipline in a strategic organizational framework with a highly competitive capacity, given its focus on efficiency and effectiveness. Likewise, it gives the discipline of projects a transversal framework of action that allows it to integrate with other disciplines in search of the common good. In this way, we can see how modern project management unashamedly addresses the frameworks of organizational management, human talent, and philosophy. Therefore, project management is losing the character of exclusivity with engineering to find complement in the organizational and human methodological approaches.

Based on these reflections, thinking of relating project management with Stoic philosophy is not a wrong position. On the contrary, when comparing some of the postulates of Stoicism with the daily reality of the project manager, many points are found in favor of strengthening the high performance of the work teams. Despite the passage of time, the Stoic philosophy remains valid and focused on achieving personal excellence through change in the way of feeling, thinking, and acting of the human being.

Now, by analyzing in detail the proposed conceptual model, the two key perspectives that positively affect high performance are distinguished: person and efficiency. This is not a new conclusion, nor is it exclusive to Stoic philosophy. The truth is that Stoicism bases its strengths on these principles. A person with a high level of emotional intelligence achieves efficiency through the quality of their decisions [45]. The success of a project depends on the high performance of the people who execute it and the control achieved over the different environmental variables.

In relation to the person perspective, we find that virtue, defined through the values that identify the person, allows us to describe human behavior and its ability to overcome

adversity, facing the challenges of the world through a positive attitude. Therefore, it can be argued that strengthening human virtue positively affects the high performance of work teams. Understanding that the mind is above our actions is perhaps one of the most significant Stoic legacies.

In the case of the variable perception of reality, stoicism breaks the premises between good and bad. The interpretation that a human being makes of his reality will lead him to make decisions that he considers to be correct, according to his mental thought process. In this sense, the objective interpretation of environmental variables is positively related to people's performance [46,47].

In relation to efficiency, focusing on the activities that are important and are under the control of the members of the work team improves the results in the management of resources. On the other hand, lifelong learning and experience gained through mistakes can stimulate high performance. Likewise, providing adequate time management allows for the prioritization of efforts by focusing on creating value [48,49].

In short and based on the validation results of the proposed hypotheses, we can conclude that the principles of Stoic philosophy positively impact the performance of work teams, allowing for the creation of value through the search for the common good as an inspiring variable for human beings [50]. The main contribution of this research is the alternative proposal described by the Stoic approach applied to project management. The novelty lies in how a contrary and classic discipline such as philosophy provides answers to current project performance problems, through a different vision for project team management. Finally, it is important to mention that the purpose of this research is to contribute to the maturity and modernization of the project management discipline.

Finally, because the authors consider the management of work teams one of the most important factors in the success of projects, we propose as future work, derived from this research, the performance of pilot tests of the results obtained in a controlled sample of project teams to monitor the results of individual and group performance.

**Author Contributions:** Formal analysis, A.S.-O.; investigation, N.M.-M., M.D.-O., A.S.-O., T.T.P.E., Y.W.N.R. and W.F.-M. All authors have read and agreed to the published version of the manuscript.

**Funding:** This research was funded by Ean University.

**Institutional Review Board Statement:** Not applicable.

**Informed Consent Statement:** Not applicable.

**Data Availability Statement:** The data were obtained from the surveyed project managers. The data are not public because it contains confidential information of the participating organizations.

**Acknowledgments:** The authors thank the project managers who participated in this study. Likewise, special recognition is given to the researchers who supported the validation of the measurement instrument. Everyone, thank you very much.

**Conflicts of Interest:** The authors declare no conflict of interest.

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
