# Peer review of "High-Performance Project Teams: Analysis from the Stoic Approach"

_sustainability, doi:10.3390/su152216095_

Round 1
Reviewer 1 Report
Comments and Suggestions for Authors
The paper is in good condition but needs to be modified to be accepted for publication. There are some comments that can help improve the quality of the paper.
1. The proposed technique must be clearly explained. The proposed method should be compared with the other state-of-the-art methods to demonstrate the efficiency of your method. You may want to better incorporate related research on the subject under study and explain against this background more compellingly what the novel contribution is that the present study intends to provide.
2. In the Introduction section, the article should present a more extensive description of the research context and what unsolved problems motivate this study.
3. On what grounds is the formulation of the two hypotheses that have been tested based in sections 4? Only on the ideas of the authors or are there published studies in which similar hypotheses are tested?
4. The paper needs much more reflection and theoretical arguments on the directions of effects and the underlying theoretical mechanisms. Beside theoretical arguments the paper here needs additional statistical checks and perhaps also test with time lag variables in a panel etc.
5. Research Methodology: Regarding the content, I think some of your statements about the methodology followed and the importance of the quantitative and qualitative tools should be backed up with some references in the field.
6. The conclusion and summary must highlight the novelty/contribution of the paper.
7. Discuss the limitations of your method and threats-to-validity of your results. Add future work in the conclusion.
8. Literature review should be more extensive. Authors are recommended to include more references and update the bibliography.
General remarks: The article must demonstrate novelty and technical contribution, with recent studies comparisons. It would be preferable that the used dataset would be extended, such that it would be generalized from solving a much too specific problem towards obtaining more generic results which would lead to increased scientific value.
Author Response
Dear Reviewer 1.
Dear reviewer, we appreciate the time to read the article and the comments. Therefore, according to your recommendations, attach the document with the explanation of the changes that were made.

Reviewer 2 Report
Comments and Suggestions for Authors
The subject proposed regarding how the postulates of the Stoic philosophy can determine the high performance of project teams it is a interesting one.
In the proposed conceptual model, the two key perspectives
that positively affect high performance are distinguished: person and efficiency and Stoicism bases its strengths on these principle.
The conclusions are clearly formulated. We suggest to present the questions of the survey (annex).
Author Response
Dear reviewer2
We appreciate the time to read the article and the comments. Therefore, according to your recommendations, attach the document with the explanation of the changes that were made.

Reviewer 3 Report
Comments and Suggestions for Authors
This is an interesting piece of work that is both multi-disciplinary in nature and is derived from a good theoretical foundation. The links between the drivers of project management from a philosophical standpoint are clear and sensibly linked and the review of smaller and larger PMOs was useful. The literature may have benefited from some further contemporary studies, but overall this was a good piece of work.
Author Response
Dear Reviewer 3.
Dear reviewer, we appreciate the time to read the article, and the comments.

Reviewer 4 Report
Comments and Suggestions for Authors
1. Content Contextualization: The content of the paper is not only succinctly described but also effectively contextualized with respect to both previous and current theoretical backgrounds. The authors have provided an exhaustive review of empirical research on the topic, making the study highly informative.
2. Relevance of Citations: All the references cited in this paper appear to be highly relevant to the research. The authors have done a commendable job in ensuring that their citations are both pertinent and up-to-date.
3. Clarity of Research Design: The research design, along with the questions, hypotheses, and methods, is stated in a clear and unambiguous manner. This makes the study easily replicable and transparent.
4. Discussion and Arguments: While the discussion section is articulate and structured, it could benefit from a more critical appraisal. Diving deeper into contrasting viewpoints or challenges in the literature can enhance its academic rigor.
5. Presentation of Empirical Results: For an empirical research paper, the results are lucidly presented with appropriate tables, figures, and necessary statistical interpretations, making it reader-friendly.
6. Referencing: The paper is adequately referenced, providing readers with an opportunity to delve deeper into particular areas of interest.
7. Conclusions and Recommendations: The conclusions, while well-supported by the results, could be more comprehensive. They, along with the recommendations, should better round out the study by capturing the essence of the findings and providing clearer future directions.
8. Quality of English Language: English language fine. No issues detected.
In conclusion, the paper titled "HIGH-PERFORMANCE PROJECT TEAMS: ANALYSIS FROM THE STOIC APPROACH" is an impressive academic piece with a fresh perspective. However, enhancing the critical assessment in the discussion and providing a more rounded conclusion will make this work even more commendable. I recommend this paper for publication with these suggested modifications.
Author Response
Dear reviewer, we appreciate the time to read the article and the comments. Therefore, according to your recommendations, attach the document with the explanation of the changes that were made.

Reviewer 5 Report
Comments and Suggestions for Authors
The paper is well structured and concise. The detailed and relevant literature review is given. The literature is contemporary.
The paper provides comprehensive research on the relationship between Stoic school principles and the high performance indicators of the project teams.
The research is clear, with hypothesis set, and research methodology well described.
The discussions and conclusions are confirming the postulates set in the hypothesis, i.e. that impacting Stoic principles, can directly lead to increase or decrease of the performance of the project teams, thus setting a solid ground on how the personal Stoic principles can be influcenced.
Author Response
Dear Reviewer 5.
Dear reviewer, we appreciate the time to read the article and the comments.

Reviewer 6 Report
Comments and Suggestions for Authors
- A more recent literature review is required.
- Discussion of the findings does not go deep enough! It needs to be strengthened.
- Be more informative in your final conclusion.
- What is your obtained contribution of doing this study? You need clearly to show it to the readers.
Comments on the Quality of English LanguageMinor checking
Author Response
Dear reviewer 6
We appreciate the time to read the article and the comments. Therefore, according to your recommendations, attach the document with the explanation of the changes that were made.

Round 2
Reviewer 1 Report
Comments and Suggestions for Authors
The revision submission is much improved and reads better. Good work!
I congratulate the authors, they have made all the modifications and from my point of view have improved the quality of the article.
I recommend publishing.
Good luck in your future work.